# Drugs, Guts, Brains, but Not Rock and Roll: The Need to Consider the Role of Gut Microbiota in Contemporary Mental Health and Wellness of Emerging Adults

**DOI:** 10.3390/ijms23126643

**Published:** 2022-06-14

**Authors:** Ju Eun Lee, David Walton, Colleen P. O’Connor, Michael Wammes, Jeremy P. Burton, Elizabeth A. Osuch

**Affiliations:** 1London Health Science Centre—Victoria Hospital, Department of Psychiatry, B8-102, London, ON N6A 5W9, Canada; jueun.lee@lhsc.on.ca; 2Rm. EC1443 School of Physical Therapy, 1201 Western Rd., London, ON N6G 1H1, Canada; dwalton5@uwo.ca; 3School of Food and Nutritional Sciences, Brescia University College, London, ON N6G 1H2, Canada; cgobert@uwo.ca; 4London Health Sciences Centre, Department of Psychiatry, Lawson Health Research Institute, Schulich School of Medicine and Dentistry, Western University, 860 Richmond Street, FEMAP, London, ON N6A 3H8, Canada; michael.wammes@lhsc.on.ca; 5Departments of Surgery, Microbiology and Immunology, Lawson Health Research Institute, Western University, London, ON N6A 3K7, Canada; jeremy.burton@lawsonresearch.com

**Keywords:** microbiota, emerging adult, adolescent, mental health/ethnology, gut–brain–microbiota axis, environmental exposure

## Abstract

Emerging adulthood (ages 18–25) is a critical period for neurobiological development and the maturation of the hypothalamic–pituitary–adrenal axis. Recent findings also suggest that a natural perturbation of the gut microbiota (GM), combined with other factors, may create a unique vulnerability during this period of life. The GM of emerging adults is thought to be simpler, less diverse, and more unstable than either younger or older people. We postulate that this plasticity in the GM suggests a role in the rising mental health issues seen in westernized societies today via the gut–brain–microbiota axis. Studies have paid particular attention to the diversity of the microbiota, the specific function and abundance of bacteria, and the production of metabolites. In this narrative review, we focus specifically on diet, physical activity/exercise, substance use, and sleep in the context of the emerging adult. We propose that this is a crucial period for establishing a stable and more resilient microbiome for optimal health into adulthood. Recommendations will be made about future research into possible behavioral adjustments that may be beneficial to endorse during this critical period to reduce the probability of a “dysbiotic” GM and the emergence and severity of mental health concerns.

## 1. Introduction: Emerging Adulthood, Mental Health, and the Gut Microbiome

Emerging adulthood (ages 18–25) is a critical period for human physical, cognitive, social, and emotional development. Post-pubertal maturations, coupled with brain neuroplasticity and the maturing activity of the hypothalamic–pituitary–adrenal (HPA) axis in emerging adults (EAs), play a vital role in neurobiological changes during this developmental period [1]. During puberty, fluctuations of the endogenous hormones such as estrogens and testosterones influence stress responses within the HPA axis, as well as brain development [2]. In the brain, changes in synaptic connectivity and axonal myelination in the frontal cortex and other regions result in changes in a sense of identity, self-consciousness, and cognitive flexibility [1,3]. Prior research also indicates that this is a window during which the onset of mental illness is most common, the overwhelming majority of which begin before the age of 24 [4].

Stress responses, including the varied activation of the HPA axis through endocrine signaling, greatly impact mental health [5]. These timely biological changes, coupled with environmental factors such as diet, exercise, and substance use, are particularly relevant to adolescents and EAs and appear to bidirectionally influence physical and psychological maturation [5]. There is a well-known bidirectional interplay between the HPA axis and the gut microbiome [6].

Consistent evidence, mostly from animal studies, shows that the gut microbiota (GM) is influenced by genetics and early-life influences such as maternal infection, mode of delivery, infant feeding, and antibiotics use, along with environmental factors such as diet, stress, childhood adversity, and exercise [7]. Westernized cultures may be particularly vulnerable during adolescent life stages, as those populations appear to be more likely to practice lifestyles that have been previously associated with disruptions of the types and diversity of GM [5]. For example, poor dietary habits, reduced physical activity, increased substance use [8], and disorganized circadian rhythms [9,10] have been more commonly associated with ‘Western’ lifestyles and many chronic illnesses—all habits that tend to start in emerging adulthood.

Recent findings suggest a unique vulnerability in the GM in the context of emerging adulthood. The GM of EAs is thought to be simpler, less diverse, and less stable than either younger or older people [5,11]. A preclinical study in mice investigating the effects of a three-week GM depletion with antibiotic treatment during adolescence and adulthood found long-lasting effects on GM composition and increased anxiety-like behavior in mice exposed during adolescence, but not in adulthood [11]. In humans, a cross-sectional study conducted in more than 1000 “ridiculously healthy” [sic] humans in China between the ages of 3 and over 100 years found that, post-weaning, the biggest perturbations in the GM occurred between the ages of 19 and 24 [12]. In these healthy people, the GM differed little after age 30 and did not decline in microbial diversity with age, though the latter finding is inconsistent across studies. Interestingly, the average GM diversity was lowest at 20 years of age [12]. In this population sample, many of the genera known for their short-chain fatty acid (SCFA) production and other metabolic capabilities of importance to the host were diminished in proportion [12]. Conversely, Zhang and colleagues found more abundant microbiota in the GM of younger people (20–30) compared to an older cohort (30–40) [13]. Collectively, these findings suggest that the EA period may be particularly vulnerable in the context of the HPA axis and sensitive to other hormonal changes, GM maturation, brain development, and the onset of mental illnesses.

According to the World Health Organization, mental illnesses cause 1 in every 5 years lived with disability, and suicide is the second leading cause of death among 15–29-year-olds [14]. We propose that emerging adulthood is a crucial period to target for making long-lasting changes in mental and physical health.

In this narrative review, we aim to focus specifically on diet, physical activity/exercise, substance use, and sleep in the context of the EA. If the GM is associated with mental health, there is evidence that the period of emerging adulthood may be critical for manipulating and establishing long-term homeostasis of the gut–brain–microbiome (GBM) axis. This leads to considering the possibility that the extensive changes in the brain during adolescence and early adulthood may function as a nexus in a series of complex, potentially bidirectional, causal relationships among genetics, lifestyle factors, mental illnesses, and the maturing GM. This is a unique lens through which to view GBM research, and we suggest that this age group (18–25) should be a separate age category in clinical trials.

## 2. The Gut–Brain–Microbiota (GBM) and the Stress Response

The GBM axis is hypothesized to function via the gut–brain neural network, the neuroendocrine–HPA axis, the gut immune system, and the neurotransmitters and neural regulators synthesized by gut bacteria [15]. Communication from the gut to the brain occurs across both the intestinal mucosal barrier and the blood–brain barrier [15]. The microbial species residing in the gut regulate the production of essential proteins and metabolites such as SCFAs, brain-derived neurotrophic factors (BDNF), and neurotransmitters such as serotonin and gamma-aminobutyric acid [16].

SCFAs, such as acetate, propionate, and butyrate, are produced by bacteria that ferment non-digestible fibers and are important metabolites in maintaining intestinal homeostasis [17]. SCFAs have various functions, including as substrates for other bacteria and a host anti-inflammatory function through the G-protein coupled receptors, as well as effects at the cellular level, such as cell proliferation, differentiation, and gene expression [17].

This has implications for the GBM axis. For example, in van de Wouw et al.’s study, when mice were exposed to acute stress, they developed changes in gut microbiota as well as behavioral and numerous physiological processes [18]. When SCFAs were administered to these mice, alterations in reward-seeking behavior, stress-responsivity, and stress-induced increases in intestinal permeability were ameliorated. Interestingly, these effects were not seen with chronic stress-induced alterations [18]. In fact, a functioning microbiota was found to be crucial for the acute stress response.

“Dysbiosis” is a vague term, but simplistically it means an imbalance in the microbiota and its collective metabolic output, which has adapted to its current environmental conditions but does not provide all the benefits for sustained homeostasis with the host. Dysbiosis may occur via changes in microbial membership or substrates through a change in diet, antimicrobial or other exposures, or genetics, amongst others. An “aberrant” or “dysbiotic” GM often increases microbial lipopolysaccharides (LPS) bacterial and other food components to permeate through the intestinal barrier more readily via epithelial cell junctions, activating gut inflammatory responses. Pro-inflammatory cytokines then stimulate the afferent vagal nerve, which has a strong influence on the HPA axis through its ascending projections to the hypothalamus [19].

Notably, microbial antigens can stimulate components of the host immune response. Through several complex pathways involving interactions between microbes in the gut and host cells, “inflammasomes”, or host cells that mediate inflammatory responses, are activated, and affect the brain via the GBM. These processes are beyond the scope of this article [20].

## 3. Associations between GM and Brain Health

Increasing evidence shows that there is a link between the GM profile and brain/mental health. However, findings between studies are not always in consensus regarding exactly what microbes are beneficial or detrimental. In preclinical rodent studies, disturbances in GM from antibiotic administration resulted in changed emotional behaviors resembling anxiety and depression [21]. In a recent systematic review of human studies, a lower abundance of *Bacteroidetes*, *Prevotellaceae*, *Faecalibacterium*, *Coprococcus*, and *Sutterella*, and a higher abundance of *Actinobacteria* and *Eggerthella* were reported in depressive disorders, and a lower abundance of *Firmicutes*, *Ruminococcaceae*, *Subdoligranulum*, and *Dialister*, and a higher abundance of *Enterobacterales* and *Enterobacteriaceae* (including *Escherichia/Shigella*) were reported in generalized anxiety disorder [16].

The GBM axis functions bidirectionally through a number of mechanisms, including via the vagus nerve, generation of metabolites such as SCFAs and enteroendocrine hormones, dysregulation of the HPA axis to alter intestinal motility, integrity, and mucus production, cross-reaction of the bacterial proteins with human antigens, and immune signaling [22]. The stimulation of immune/inflammatory pathways further reveals a potentially important link between gut “dysbiosis” and the current inflammatory theory of depression in humans [23]. Additionally, it plays a role in the development of other neurological and psychiatric diseases such as Parkinson’s disease (PD), Alzheimer’s disease, multiple sclerosis, and autism spectrum disorder [20]. For example, in PD, there is increased neuroinflammation, dopaminergic neuronal death, and α-synuclein in the brain, and increased intestinal permeability and dysbiosis in the gut. Via the mechanisms of the GBM axis, i.e., the vagal nerve, the CNS and the enteric nervous system bidirectionally perpetuate immune dysregulation, inflammation, and cytokine increase, further increasing the severity of PD [22]. The mechanisms and the function of the GBM axis in the developmental of neurological and psychiatric illnesses remain complex and multifactorial and require further research and understanding the pathophysiology of the GBM axis.

As mentioned, evidence indicates that modulation of the GM may be both more widespread and more impactful during EA life stage than at other life stages. If, indeed, an optimal period for developing a healthy microbiota can be identified, this may provide an opportunity to prevent mental illnesses that emerging evidence indicates are influenced by the GBM, such as depression and anxiety [24]. Additionally, it is possible that GM-modifying interventions may be useful as independent or adjunctive therapy alongside pharmacotherapy and psychotherapy interventions for enhanced effect.

Conversely, the creation of “dysbiosis” during the time of emerging adulthood could result in a cascade of effects that have long-lasting negative influences on health overall as well as mental health. Below, we focus on emerging adulthood as the potential window in which to establish reduced opportunities for the disruption of GBM homeostasis. We will review some of the most salient factors that could be disrupting EAs’ GBM and affecting their mental (and physical) health. These include diet, physical exercise, substance use, and sleep (Figure 1).

## 4. Lifestyle Factors Affecting GM and Mental Health: Diet, Physical Activity/Exercise, Substance Use, Sleep 

### 4.1. Diet

Diet in EAs in Western societies can be affected by various factors. Common barriers to healthy eating include time constraints, unhealthy snacking, convenience high-calorie food, stress, high prices of healthy food, easy access to junk food [25], and food insecurity [26]. A Western diet, which refers to the excessive consumption of simple sugars, saturated fats, animal proteins, and reduced fruit and vegetable fibers, is particularly associated with adverse health outcomes [27]. For adolescents and young adults, the Western diet is associated with periodontal disease [28], negative mental health outcomes [29], obesity, and cardiometabolic risks [30]. It is important to consider the impact of unhealthy dietary habits on the GM in EAs.

The Western diet is associated with reduced *Bacteroides*, *Verrucomicrobia*, *Eubacterium rectale*, *Clostridium coccoides*, and *Bifidobacterium* [27], increased *Firmicutes*, *Proteobacteria*, *Clostridiales*, *Ruminococcaceae*, *Bacteroidales*, and *Prevotellaceae*, and lower production of SCFAs [31,32]. These are associated with negative health outcomes such as the development of metabolic diseases and neuropsychiatric consequences such as anxiety [27,32]. In preclinical models, high-fat diets have been associated with triggering microbial dysbiosis, intestinal permeability, and inflammation in the GM, such as by the depletion of acetate-producing *Bifidobacteria* [33]. This is further associated with susceptibility to infections [34] and pro-inflammatory cytokines such as interleukin (IL)-1, IL-6, and tumor necrosis factor (TNF)-α [27].

Healthier diets improve the GM composition and activity. For example, the Mediterranean diet consists of a high intake of fruit/vegetables/nuts, moderate consumption of fish, and low intake of saturated fat, meat, and dairy products [35] (Figure 2). It is associated with increased bifidobacterial counts, increased concentrations of SCFA, and a lower *Firmicutes*/*Bacteroidetes* ratio [36]. Dietary fiber, plant-based foods, vegetables, and fermented foods are associated with increased production of anti-inflammatory SCFA [33], and micronutrients such as polyphenols and omega-3 fatty acids are associated with increased *Lactobacillus* and *Bifidobacteria* at the gastrointestinal level [37].

A reduced food intake, as seen in individuals with anorexia nervosa (AN), also results in an altered GM and body-wide alterations [38]. For example, transplantation of stool from individuals with AN to germ-free mice resulted in reduced appetite, lower weight gain, less energy use, and even obsessive-compulsive and anxious behaviors, characteristics found in humans with AN [39]. The GM in patients with AN is thought to have different energy-extracting capabilities, which explains the greater caloric requirements to gain weight [38,39].

Food directly alters the microbiota, but the microbiota also affects the nutritional value and metabolism of food [34]. Preclinical studies have shown that GM regulates and modulates energy extraction from food along with the regulation of energy expenditure, appetite and satiety, glucose homeostasis, and lipid metabolism [40].

The GBM axis suggests multiple ways in which associations between poor diet and mental illness exist, such as HPA axis hyperactivation and secretion of glucocorticoids, reduced hippocampal neurogenesis, altered kynurenine pathway, mitochondrial dysfunction, reduced serotonin and dopamine, and release of inflammatory cytokines [32]. Certain foods and micronutrients such as polyphenol-rich dark chocolates and vitamin C have been shown to decrease cortisol levels [32], and antioxidant properties of certain nutrients such as omega-3 fatty acids have been found to mitigate inflammation-induced reductions in neurogenesis [41]. Conversely, high amounts of sugar and fat in the diet are associated with impaired neurogenesis and reduced BDNF levels within the hippocampus, thereby impacting cognitive performance [42].

Multiple studies have found associations between diet and mood/anxiety symptoms [32,43,44]. For example, a randomized control trial (RCT) of young adults (aged 17–35) with elevated levels of depression and poor diet undergoing a brief three-week diet intervention compared to habitual diet found that the diet group reported lower depression symptoms than controls [45].

There is also increasing evidence for the role of prebiotic and probiotic supplementation to improve health outcomes. *Akkermansia muciniphila* is one of many studied supplements, with its positive association to metabolic health [46]. It is also a strain of interest in mental health, with associations between metabolic/inflammatory diseases and mental illness, as well as for the weight gain and metabolic side effects of psychotropic medications. Psychobiotics, which refers to prebiotics and probiotics that influence the central nervous system (CNS) by the GBM axis, is an emerging option of treatment for neuropsychiatric symptoms [47]. Psychobiotics influence the CNS via the vagus nerve through the action of SCFAs, enteroendocrine hormones, cytokines, and neurotransmitters [47]. Preclinical studies show positive mental health outcomes; however, individual differences (i.e., sex, gender, diet), different types of strains in the supplements, long-term investigations, dosage, and good-quality clinical data are current barriers, and remaining questions need to be answered before they are more widely disseminated and practiced [47].

Dietary interventions may be especially effective for EAs. For example, diet interventions may serve as a behavioral activation intervention, which is part of a well-established approach to managing depression in EAs [48]. If our hypothesis is true and the GM of EAs are more unstable and susceptible to change, then brief diet interventions such as those found in Francis et al.’s work [45] may lead to significant changes in the GM and mental health of EAs.

**Figure 2 ijms-23-06643-f002:**
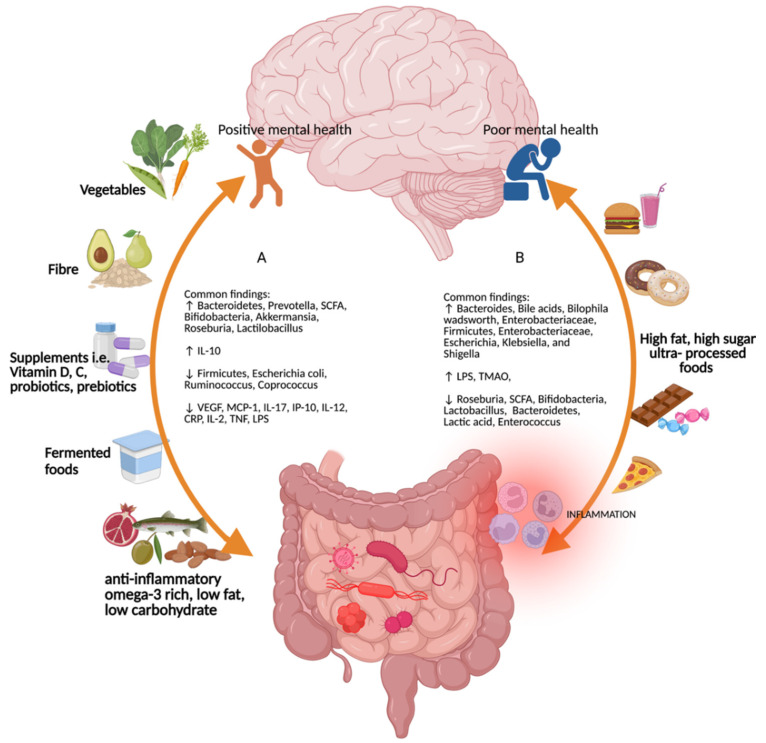
Common findings for the different types of diet on the gut–brain–microbiome axis. (**A**) Diets rich in vegetables, fiber, micronutrients such as vitamins D and C, probiotics and prebiotics, fermented foods, anti-inflammatory omega-3-rich, low-fat, and low-carbohydrate foods promote positive mental health and increases in *Bacteroidetes*, *Prevotella*, short-chain fatty acids, *Bifodobacteria*, *Akkermansia*, *Roseburia*, *Lactilobacillus*, and interleukin (IL)-10, and decreases in *Firmicutes*, *Escherichia coli*, *Ruminococcus*, *Coprococcus*, vascular endothelial growth factor, monocyte chemoattractant protein-1, interferon gamma-induced protein 10, IL-17, IL-12, c-reactive protein, IL-2, tumor necrosis factor, and lipopolysaccharide [49,50,51]. (**B**) High-fat, high-sugar, and ultra-processed foods increase *Bacteroides*, bile acids, *Bilophila wadsworth*, *Enterobacteriaceae*, *Firmicutes*, *Enterobacteriaceae*, *Escherichia*, *Klebsiella*, and *Shigella* [49,52]. Figure created with Biorender (accessed on 29 April 2022).

### 4.2. Physical Activity and Exercise

Physical activity and exercise are well-known, effective strategies for improving physical and mental health. Exercise is of particular importance among EAs, with its influence on a variety of systems, including neuroendocrine, neurogenesis, oxidative stress, autoimmune, and cortical structural changes [53], as well as on HPA axis regulation [54]. Exercise in EAs is associated with improvements in adiposity levels, blood pressure, plasma lipid and lipoprotein levels, non-traditional cardiovascular risk factors, as well as mental health characteristics such as self-concept, anxiety, and depression [55]. Additionally, fitness in early adulthood (ages 18–30) is strongly associated with improved long-term mortality independent of obesity or other indices of metabolic or cardiovascular risk [56].

Physical activity is important for GM diversity and the regulation of health-beneficial microbes and metabolites. Exercise has a role not only in altering the GM diversity but in the production of bioactive metabolites such as butyrate [57], as the fermentation of SCFAs are dependent on factors such as gut peristalsis, which is affected by exercise [58]. Exercise is thought to increase gut motility, thereby increasing the availability of carbohydrates and amino acids in the distal colon for increased bacterial metabolic processes and fermentation efficiency [58]. Lactate produced by skeletal muscle energy metabolism crosses the epithelial barrier into the intestinal lumen and may also contribute as a carbon source for certain SCFA-producing bacteria such as *Veillonella*, and to produce propionate, which enhances energy production and prevents the accumulation of lactate [59].

Several preclinical studies have found an increase in *Akkermansia muciniphila* and *Fecalibacterium* associated with physical exercise [60,61,62,63,64,65]. However, findings are inconsistent across studies due to participant variability in gender, geography, baseline body mass index (BMI)/activity level, diet, age, genetics, and the exercise modalities used [58]. For example, in Allen et al.’s study, after a six-week exercise intervention, the abundance of butyrate-producing taxa and fecal acetate and butyrate concentrations were only increased in lean subjects but not in obese subjects [62]. This suggests that there may be functional differences depending on a variety of factors, such as BMI, genetics, and diet.

Mika et al. found that in juvenile rats, with their less even and diverse GMs, exercise had a greater impact on the GM than in adult rats and produced patterns associated with adaptive metabolic consequences [66]. This included increased SCFA production, increased energy expenditure, inhibited fat accumulation in adipose tissue, as well as a greater relative abundance of Bacteroidetes and decreased *Firmicutes*, all reflective of a lean phenotype. Nevertheless, a recent 12-week RCT on aerobic (moderate-intensity) exercise on GM in adolescent humans (aged 12–14) with subthreshold mood syndromes found no significant changes in the GM [67].

A few important findings stand out in the literature regarding exercise and the GM. The intensity of exercise is important, and the optimal level may need to be individualized. Intense exercise can be inflammation-promoting [68] and can lead to dysbiosis via increased heat stress, reduction of intestinal blood flow and gut ischemia, and transient impairment of gut barrier function [69,70,71]. This is of significant interest to the EA population, particularly for young athletes and for those EAs who use excessive exercise to manage body image. 

Thus, there are individual differences based on age, gender, genetics, baseline BMI, baseline activity level, and diet that are important. Exercise alone may not independently alter the GM of humans. For example, diet, which often changes with changes in activity levels, may be an important co-factor in determining the benefits of exercise on individuals.

### 4.3. Substance Use

Emerging adulthood is a period of increased risk-taking behaviors, including with substance use. Substance use among EAs is increasing with the increased availability and popularity [72,73,74]. The use of electronic cigarettes (E-cigarettes/vaping) has increased in popularity in youth [75]. Alcohol use remains consistently popular among EAs and there has been an increase in cannabis use in recent years, particularly with legalization in various Western countries including Canada, parts of the United States, and Europe. In this section, nicotine, alcohol, and cannabis effects on the GM of EAs will be discussed.

#### 4.3.1. Nicotine

E-cigarettes have been the most commonly used tobacco product among youth since 2014 [75]. One in nine high school students reported using E-cigarettes in 2021 in the United States [76], and one in four of European youth (aged 15–24) reported trying E-cigarettes in a 2020 report [77].

The neuronal nicotinic acetylcholine receptors are a key factor in brain development and are highly critical for the formation and the maturation of the central nervous system (CNS) [78]. Early nicotine use affects impulsivity and addiction and can potentiate the development of psychiatric disorders [79,80]. Chronic nicotine exposure during adolescence exerts long-term effects on cognitive processing, including attention [80].

The underlying mechanisms of nicotine effects on GM are largely unknown but are thought to involve increased permeability of intestinal mucosa and impaired mucosal immune responses [81]. These may occur via altered levels of SCFA and bile acids [82,83]. Nicotine use is associated with significantly lower bacterial diversity in the upper small intestinal mucosa [82]. However, conflicting microbial findings are reported in the literature on smoking and the GM [16]. For example, Shanahan et al. found increased relative abundance of *Firmcutes* and *Actinobacteria*, with lower levels of *Bacteroidetes* (*Prevotella*) and *Proteobacteria* (*Neisseria*) [84]. Other studies have found an increase in *Proteobacteria* and *Bacteroidetes*, as well as *Clostridia* and *Prevotella*, in smokers [82]. The effects of nicotine on the GM may depend on the mode of intake and individual factors such as sex, diet, and genetics [82].

#### 4.3.2. Alcohol

Alcohol use remains high among EAs. The 2019 Youth Risk Behavior Survey data found that 29% drank alcohol and 14% binge drank during the past 30 days in the United States [85]. In Europe, approximately 37% of 15-year-olds used alcohol [86]. Alcohol use in EAs is commonly in the form of heavy and binge drinking [87]. However, alcohol dependence does occur, especially in those with predisposing factors such as mental illness and family history of substance use, and vulnerable social and environmental factors such as adverse childhood experiences [88,89,90]. Binge and heavy drinking in adolescents are associated with changes in frontal and temporal lobes, as well as interconnecting networks associated with learning, memory, visuospatial functioning, psychomotor speed, attention, executive functioning, and impulsivity [91].

Alcohol consumption is associated with reduced *Bifidobacteria*, *Bacteroidetes*, and *Lactobacillus*, and increased pro-inflammatory *Proteobacteria* in the gut [82]. Changes in the GM, along with an impaired intestinal barrier and mucosal inflammation secondary to excessive alcohol use, are thought to play a key role in the development and maintenance of alcohol use disorders as well as alcohol-related liver disease [82]. Alcohol induces microbial changes with significant variability in findings due to the types of alcohol ingested, quantity and duration of use, comorbid use of other substances, and individual differences (genetics, comorbid illness). In general, alterations in the metabolites of GM by alcohol, such as bile acids, have been identified to increase inflammation and decrease intestinal epithelial integrity [82].

#### 4.3.3. Cannabis

In Canada, cannabis use among EAs is double that of individuals aged 25 years old, and 31% of EAs have reported an increase in cannabis consumption due to the COVID-19 pandemic [92]. In Europe, 20% of young adults (aged 15–24) reported using cannabis in the past year [93], and in the United States, 22% of high school students reported use of cannabis in the past 30 days in 2019 [94]. Cannabis use has been associated with deficits in attention, memory, processing speed, visuospatial functioning, planning, and sequencing ability [95,96]. These effects are more severe and longer lasting with earlier onset of use and higher quantity used [96].

The endocannabinoid system is thought to play an essential role in the interactions within the GBM axis. The cannabinoid 1 receptor (CB1R) is highly expressed in the intestinal epithelium, smooth muscle, the submucosal myenteric plexus, and in the brain [82]. The cannabinoid 2 receptor (CB2R) is expressed more frequently in the plasma cells, macrophages, and the microglial cells of the brain [82]. Tetrahydrocannabinol binds mainly to the CB1Rs, while cannabidiol has a higher affinity for CB2Rs. Activation of the CB1Rs reduces intestinal motility, gastric acid secretion, and nausea, and enhances food intake [82]. CB2R modulates inflammatory mechanisms by altering the integrity of the intestinal epithelial barrier, thereby affecting gut epithelial permeability [97].

In one preclinical study, the agonism of cannabinoid receptors showed increased plasma levels of pro-inflammatory LPS and changes in mRNA expression of proteins such as occludin and zonulin, which are involved in modulating intestinal endothelial barrier permeability [98]. By antagonizing the CB1R, the authors then reduced the levels of LPS and improved intestinal barrier permeability [98]. Similar findings were reported when probiotics (live microorganisms of benefit to the host) were administered to antibiotic-treated mice: treatment with probiotics induced changes in the expression of mRNA for the CB1R receptor in the colon [97]. These findings suggest potential for changes in the GM from alterations in the endocannabinoid system by environmental exposure to cannabis, although this remains to be demonstrated in humans.

Panee et al. conducted a pilot study in humans that compared mitochondrial function, microbiota composition, and five domains of cognitive function in chronic versus non-cannabis users [99]. They found that lifetime cannabis use was inversely correlated with the *Prevetolla*/*Bacterioides* ratio, mitochondrial function, and the inhibitory control and attention subdomain of cognitive function. Associations among microbiota, mitochondrial function, and cognition were not observed in non-cannabis users. With advances in understanding the GBM and cannabinoid and other receptors, the targets for microbiota therapy are shifting towards the regulation of endocannabinoid precursors. It is important to consider cannabis use by EAs and its effects on physiology and the impact on establishing long-term properties of the GM and GBM.

### 4.4. Sleep/Circadian Rhythm

Circadian rhythm refers to the complex endocrine and molecular processes together with resulting behavioral, biochemical, and physiological functions that occur over a period of 24 h. It is modulated by the suprachiasmatic nucleus (SCN) in the hypothalamus, which receives innervation from the eyes via the retinohypothalamic tract [100]. The function of this human biological clock is influenced by the light–dark cycle, along with signals such as diet, exercise, temperature, and infection [101]. Some of the disruptions to the circadian rhythm that occur in modern society include shift work, jet lag, inconsistent eating times, “Western” high-fat diet, and night-time light exposure or food intake [101].

Circadian dysrhythmia is linked to mental illnesses such as depressive and anxiety disorders [102]. Seasonal affective disorder (SAD) (DSM-V: MDD with seasonal pattern) is a depressive disorder with symptoms typically during the fall and winter months when there is less sunlight. The pathophysiology of SAD is thought to be related to circadian rhythm disruption, dysregulation of the melanopsin signaling pathway and its impact on the serotonin reuptake, and the dysfunction of the HPA axis [103]. Circadian rhythm interruptions may therefore be more common and impactful during emerging adulthood, when disruptions occur both intrinsically via hormonal and physiological changes, and extrinsically via socially driven behavioral alterations.

During puberty, there is a shift in the circadian rhythm, with a delay in the timing of sleep, called delayed circadian phase [104], as well as lower melatonin secretion amplitude, and a decline in delta and theta non-rapid eye movement sleep [105]. On average, there is a sleep delay of 1–3 h, greater in boys than in girls, with the age of peak delay at 15–21 years, and the age of onset of the delay with the onset of puberty [106]. The HPA axis also has a strong diurnal rhythm, with high levels of endogenous cortisol upon awakening and the lowest around midnight. This rhythm coincides with the sleep–wake cycle along with the SCN, which also regulates the periodicity of corticosteroid activity [107].

The GM also functions under a diurnal rhythm and may be influenced by circadian disruptions [101,108]. For example, in Bowers et al.’s study, sleep disruptions were associated with an increase in the *Firmicutes*/*Bacteroidetes* ratio, decreases in the genus *Lactobacillus*, phylum *Actinobacteria*, genus *Bifidobacterium*, and reduced levels of bile acids [109].

Many bacterial genera and species also exhibit oscillatory behavior in response to time of eating [101]. For example, *Bacteroidetes* genera and *Clostridia* species both oscillate during light–dark cycles in mice, with different abundances of bacteria during the active versus rest phases [110]. Where diurnal oscillations were disrupted by deleting circadian clock genes, altering the timing of or restricting food availability, or changing the light–dark phase, the gut microbial diurnal rhythmicity was also disrupted [111]. The timing of food intake—chrono-nutrition—may therefore be an important factor in the regulation of the GM. In Berendsen et al.’s study, adolescents (aged 13–20) with delayed sleep phase disorder consumed dinner more regularly, had snacks in the morning less frequently, and consumed their first food of the day later than controls [112], suggesting the importance of chrono-nutrition.

## 5. Discussion

Although challenges exist in finding high-quality controlled human studies related to the GM and how it relates to mental health in EAs, there is enough evidence to suggest that this life stage is characterized by biological and environmental vulnerabilities within both the brain and the gut. Effects of exposures to conditions that impact GM during this stage could have broad impacts on health, including mental health, for the duration of the individual’s life. This has vast healthcare implications.

GM composition studies of humans often have very different participant profiles and have poor generalizability across populations. In the context of EAs, the confounders may be even more complex given the overall instability in factors such as diet, substance use, hormonal changes, circadian variability, and stress. In the context of future research, EAs are currently more often classified as part of the adult cohort (ages 18–65) and are not often separately analyzed. As a result, unique findings may be missed from this group in research involving the GM, GBM, neurophysiology, and psychiatry. Appropriate grouping of ages is therefore recommended, with special attention to the vulnerabilities during the EA period.

Evidence suggests that the impact of diet, physical activity/exercise, substance use, and sleep on the GBM axis and EAs’ mental health are strong despite inconsistent findings on the GM. The various hormonal and environmental stressors during emerging adulthood influence one another and synergistically impact EAs’ health. For example, in addition to examining the “dysbiosis” or a maladapted microbiota from any specific substance use, understanding the changes in diet independent of or secondary to the individual substance use or how macro- and micro-nutrients are absorbed by the GM is a potentially interesting and important topic to explore.

The food guidelines generally target all ages, with specific caloric and micronutrient recommendations for certain age groups, such as children. To our knowledge, there is no specific recommended diet for EAs. Given the benefits of certain foods on the GM, such as fermented foods [113], future research is recommended to understand the short- and long-term impact of healthy GM-promoting nutrients for this age group. Furthermore, in the context of emerging adulthood, researchers need to understand the relationship between food, activity/exercise, substance use, and sleep, and their short- and long-term impact on the GM of EAs.

Due to the early preclinical findings described here, scientists are working to identify potentially beneficial microbes and administrative mechanisms to promote health and to administer in the context of specific diseases to alter the GM environment and thereby either prevent or alleviate clinical symptoms. While this may seem overly ambitious given the individual non-modifiable differences in human populations, variable environmental factors, and perhaps a narrow window of opportunity to treat conditions, the opportunities offered by personalized, microbiota-focused interventions in disease are compelling.

Mental illnesses commonly arise in emerging adulthood, and their symptoms can further compromise the gut health via poor diet, reduced activity, substance use (including medications), and poor-quality sleep. Interventions to alleviate symptoms or to prevent psychiatric illnesses as primary prevention are generally still quite limited. Interventions are currently most often restricted to secondary and tertiary interventions such as pharmacotherapy and psychotherapy. While newer approaches are being explored (e.g., brain stimulation [114] and psychedelic therapies [115]), convergent lines of evidence lead us to raise the possibility of a role for microbial therapeutics as primary and/or secondary interventions for mental illnesses. With the emerging research and evidence for fecal material transplant, pre-, pro-, and post-biotics to promote human brain health through their effects on gut health, psychobiotics may become a new potentially cost-effective treatment and/or preventative gut- and mind-altering therapy. Psychological and societal/public health long-term approaches to the reduction and management of poor mental health for EAs should involve psychoeducation on healthy lifestyle behaviors to enhance GM health, such as good nutrition, exercise, substance abstinence, and proper sleep.

An important consideration not discussed in this paper is the use of prescription medications such as antidepressants, antipsychotics, and stimulants for psychiatric illness among EAs. With the increasing prevalence of mental illnesses among EAs and medication use as a common, convenient, and often economical treatment, it is likely that psychotropic medication use will increase over time [116]. However, medication switches and poor adherence among EAs are barriers to treatment in this group [117], and some psychiatric conditions and medications are associated with cardiometabolic syndrome [118,119,120]. Alterations in the GM secondary to various psychotropics may be an important consideration for future analysis of the GM of EAs [121,122].

## 6. Conclusions

While it is currently unclear which combination of factors are most detrimental to mental health with regards to the interaction with the GM, we know that healthy diet and lifestyle with the least interference from chemicals with antimicrobial properties are associated with the best health outcomes. Disadvantageously functioning homeostasis of GM and GBM in the vulnerable years of emerging adulthood may lead to chronic and/or lifelong challenges to health and mental health. Given the environmental and societal forces at play during this stage of life, special attention and intervention in this age group could have a great impact at both the individual and societal levels.

## Figures and Tables

**Figure 1 ijms-23-06643-f001:**
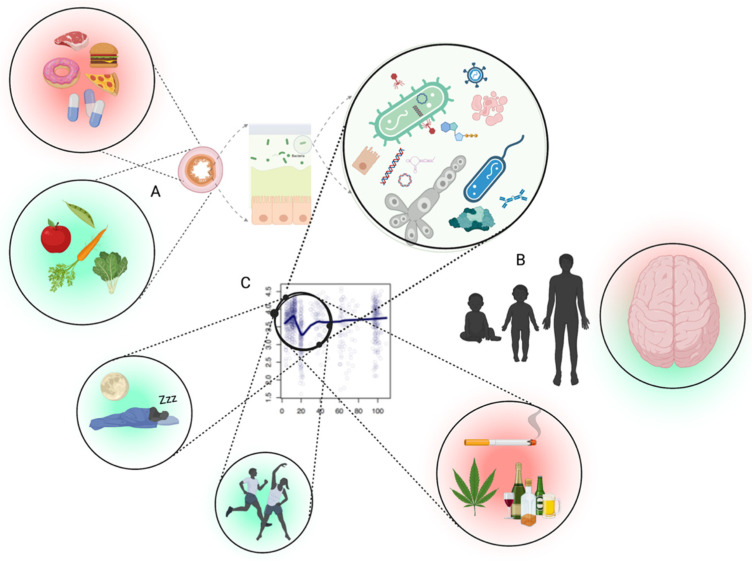
Interface between the gut microbiota and mental health likely depends upon several factors. (**A**) The first being the inputs to the intestinal tract which shape the microbiota accordingly (diet, medication, antimicrobials, etc.). (**B**) Periods where microbiota goes through changes in diversity (alpha) occur in healthy people, especially between late teens and early twenties, which likely result in differences of metabolic output which impact brain health. (**C**) The intersection of the adolescent brain, combined with a normally fluctuating microbiota of the age group, the promotion of a desirable microbiota through physical activity/exercise and circadian rhythm, and less desirable microbiota using different substances. Part (**C**) adapted from Bian et al., 2017 [12]. Figure created with Biorender (accessed on 29 April 2022).

## Data Availability

Not applicable.

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
