# Peer review of "Drugs, Guts, Brains, but Not Rock and Roll: The Need to Consider the Role of Gut Microbiota in Contemporary Mental Health and Wellness of Emerging Adults"

_ijms, 2022, doi:10.3390/ijms23126643_

Round 1
Reviewer 1 Report
The authors focused on a Review: Drugs, guts, brains, but not rock and roll: The need to consider the role of gut microbiota in contemporary mental health and wellness of emerging adults. Please see few of my main concerns/suggestions regarding this manuscript:
- Main text must be set as Justified.
- At the final of Introduction section, the aim of the study is missing. What makes special this study? Which is its novelty character or its special aspects? Why have the author chosen this topic? What differentiate this paper from others already published in the same/similar topic? Numerous studies are done in this field, so it is important to emphasize what does your study brings special.
- I suggest a new Section Methods. As the authors have stated that this is a Review, a PRISMA flow chartis recommended. I suggest checking Page MJ, McKenzie JE, Bossuyt PM, Boutron I, Hoffmann TC, Mulrow CD, et al. The PRISMA 2020 statement: an updated guideline for reporting systematic reviews. BMJ 2021;372:n71. doi: 10.1136/bmj.n71 regarding PRISMA flow chart. Please take care and detail in the best way the inclusion/exclusion criteria used for the literature selection. Include here also the MeSH terms. Do not forget renumbering the following sections.
- Latin names must be written in Italics, respecting the international regulation in this regard.
- Section 3. Please detail the Dysregulation process of the Gut-Brain Axis, Dysbiosis and Influence of numerous other factors on Gut Microbiota associated to neurodegenerative diseases. I suggest checking and referring to https://doi.org/10.2174/1570159X18666200606233050
- Below each Figure (i.e. Figure 2), please detail each abbreviation used on the figure. Abbreviations must also respect the instructions for authors: “Acronyms/Abbreviations/Initialisms have been defined the first time they appear in each of three sections: the abstract; the main text; under the first figure or table. When defined for the first time, the acronym/abbreviation/initialism should be added in parentheses after the written-out form”. Please check and revise the entire manuscript in this regard.
- I suggest detailing/completing the paragraph with the role of supplements (pre/probiotics) https://doi.org/10.3390/microorganisms9030618 as it is well known the current trend of supplementing the diet with various nutritional supplements.
- Last section Conclusions is missing. I suggest Conclusions and future directions (or something similar). It is very relevant to have a complex overview regarding the main directions in this topic.
Reviewer 2 Report
The manuscript relates to an important issue of emerging adults mental health and the main factors on which it depends. As we observe a gradual increase in mental problems among societies, including people from this age group, the issues of mental health protection in emerging adults should be widely subject to research and discussion. The authors note the importance of proper microbiota as the key to mental health. It seems that the diet has the greatest influence on the composition and abundance of microbiota in the long term. If emerging adults show specific conditions for the growth of microbiota, due to the level of hormones and other factors, should the proper diet of people in this period of life differ in any way from the diet of people in other age groups, what exactly should it contain? Is it enough not to use stimulants? Please respond to these questions in the manuscript.
Round 2
Reviewer 1 Report
The authors responded to my suggestions.